# Evolutionary emergence of Hairless as a novel component of the Notch signaling pathway

Steven W Miller[1], Artem Movsesyan[1], Sui Zhang[1], Rosa Fernández[2†], James W Posakony[1]*

[1]Division of Biological Sciences, Section of Cell and Developmental Biology, University of California, San Diego, La Jolla, United States; [2]Bioinformatics and Genomics Unit, Center for Genomic Regulation, Barcelona, Spain

**Abstract** Suppressor of Hairless [Su(H)], the transcription factor at the end of the Notch pathway in *Drosophila*, utilizes the Hairless protein to recruit two co-repressors, Groucho (Gro) and C-terminal Binding Protein (CtBP), indirectly. Hairless is present only in the Pancrustacea, raising the question of how Su(H) in other protostomes gains repressive function. We show that Su(H) from a wide array of arthropods, molluscs, and annelids includes motifs that directly bind Gro and CtBP; thus, direct co-repressor recruitment is ancestral in the protostomes. How did Hairless come to replace this ancestral paradigm? Our discovery of a protein (S-CAP) in Myriapods and Chelicerates that contains a motif similar to the Su(H)-binding domain in Hairless has revealed a likely evolutionary connection between Hairless and Metastasis-associated (MTA) protein, a component of the NuRD complex. Sequence comparison and widely conserved microsynteny suggest that S-CAP and Hairless arose from a tandem duplication of an ancestral MTA gene.
DOI: https://doi.org/10.7554/eLife.48115.001

*For correspondence: jposakony@ucsd.edu

Present address: †Department of Life Sciences, Barcelona Supercomputing Center, Barcelona, Spain

Competing interests: The authors declare that no competing interests exist.

## Introduction

A very common paradigm in the regulation of animal development is that DNA-binding transcriptional repressors bear defined amino acid sequence motifs that permit them to recruit, by direct interaction, one or more common co-repressor proteins that are responsible for conferring repressive activity. Two such universal co-repressors are Groucho (Gro) and C-terminal Binding Protein (CtBP).

The ancient and highly conserved transcription factor Suppressor of Hairless [Su(H)] functions at the terminus of the widely utilized Notch cell-cell signaling pathway. Su(H) is converted into an activator by signaling through the Notch receptor, but in the absence of signaling it functions as a repressor. Earlier studies have revealed that in many settings in *Drosophila*, Su(H)'s repressive activity depends on binding to the Hairless protein (*Figure 1*). Hairless includes separate Gro- and CtBP-binding motifs, which permit it to function as an adaptor to bring these two corepressors to Su(H) (*Figure 1B*) (*Barolo et al., 2002a*). Thus, the Su(H)/H partnership in the fly represents a notable exception to the rule of direct co-repressor recruitment.

As genome and transcriptome sequences have become available for more and more insects and other arthropods, we have searched for possible Hairless orthologs in a wide variety of species, in an attempt to determine the protein's phylogenetic distribution. We have found that Hairless is confined to the Pancrustacea (or Tetraconata), a clade of arthropods that includes the Crustacea and Hexapoda (*Misof et al., 2014*; *Kjer et al., 2016*). While this indicates that Hairless was gained at least 500 Mya, it also raises the question of how Su(H) in other protostomes acquires repressive activity.

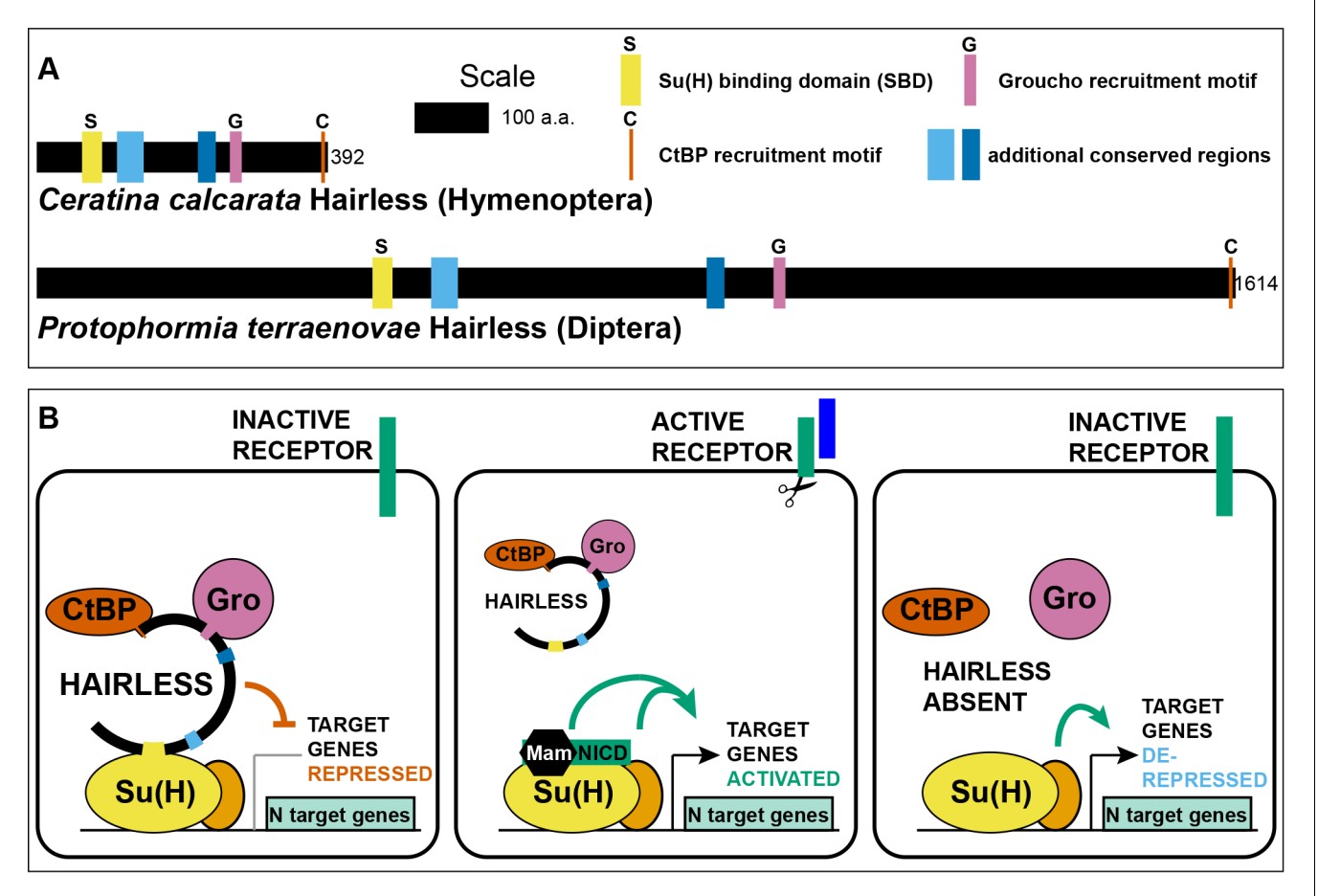

**Figure 1.** Hairless mediates indirect recruitment of co-repressor proteins to Su(H). (**A**) Diagram denoting locations of conserved domains and motifs within Hairless, and illustrating extreme size differences of the protein in different species. Shown is Hairless from the carpenter bee *Ceratina calcarata* and the blowfly *Protophormia terraenovae* (**Hase et al., 2017**), with scale and protein sizes indicated. (**B**) Summary of Hairless's known mode of action (**Lai, 2002**; **Maier, 2006**) as an adaptor protein that recruits the global co-repressors C-terminal Binding Protein (CtBP) and Groucho (Gro) to Suppressor of Hairless [Su(H)], the transducing transcription factor for the Notch (N) cell-cell signaling pathway; adapted from Figure 6 of **Barolo et al. (2002a)**. In the absence of signaling through the Notch receptor (left), Su(H) acts as a repressor of Notch target genes, despite the presence of transcriptional activator proteins (orange oval). Upon activation of the Notch receptor (middle), Su(H), in a complex with the receptor's intracellular domain (NICD) and the co-activator Mastermind (Mam), functions to activate transcription of pathway target genes in cooperation with other transcriptional activators. In the absence of Hairless and hence in the absence of Su(H)'s repressive activity (right), the partner transcription factors are often sufficient to activate expression of target genes in a signal-independent manner (**Barolo and Posakony, 2002b**).
DOI: https://doi.org/10.7554/eLife.48115.002

The following figure supplement is available for figure 1:

**Figure supplement 1.** Graph showing predicted disordered regions in *Drosophila melanogaster* Hairless, generated by DISOPRED3 (**Buchan et al., 2013**; **Jones and Cozzetto, 2015**).
DOI: https://doi.org/10.7554/eLife.48115.003

Here we present evidence that direct co-repressor recruitment by Su(H) is likely to be ancestral in the protostomes. We show that Su(H) in a broad range of protostomes, including arthropods, molluscs, and annelids, bears both a short linear motif that mediates binding of CtBP and a novel motif for direct recruitment of Gro. Thus, the evolutionary appearance of Hairless has permitted the replacement of an ancient and predominant regulatory mechanism (direct co-repressor recruitment) with a novel one (indirect recruitment).

What can we learn about the evolutionary history of Hairless? While Hairless itself is found only in the Pancrustacea, we show that the genomes of Myriapods and Chelicerates encode a protein with clear sequence and functional similarities to Hairless. These proteins include a motif that strongly

resembles the Su(H)-binding domain of Hairless, and we demonstrate that this motif from the house spider *Parasteatoda tepidariorum* does indeed bind Su(H). In addition, these Myriapod and Chelicerate proteins also include one or more canonical motifs for recruitment of CtBP. Accordingly, we designate these factors as 'Su(H)-Co-repressor Adaptor Proteins' (S-CAPs).

Finally, further sequence analyses, along with the discovery of conserved microsynteny, have provided substantial evidence that Hairless and the S-CAPs are likely to be homologous and that they arose from a duplication of the gene encoding Metastasis-associated (MTA) protein, a component of the nucleosome remodeling and deacetylase (NuRD) complex.

An intriguing question in evolutionary biology concerns the path by which a particular clade has escaped a strongly selected character that has been conserved for hundreds of millions of years. We believe that our study has yielded valuable insight into both the emergence of an evolutionary novelty and its replacement of an ancestral paradigm.

## Results

### Hairless is present only in the Pancrustacea

We have conducted extensive BLAST searches of genome and transcriptome sequence data for a wide variety of metazoa in an attempt to define the phylogenetic distribution of Hairless. We find that Hairless as originally described (*Bang and Posakony, 1992*; *Maier et al., 1992*; *Maier et al., 2008*) is confined to the Pancrustacea (or Tetraconata), and occurs widely within this clade, including the Hexapoda, Vericrustacea, and Oligostraca (*Figure 2A*). By contrast, no evidence for a true *Hairless* gene has been detected in either Myriapods or Chelicerates, even in cases where substantially complete genome sequence assemblies are available.

The enormous variation in the size of the Hairless protein in various Pancrustacean clades is worthy of note (*Figure 1A*). The known extremes are represented by the Diplostracan (shrimp) *Eulimnadia texana* (343 aa) (*Baldwin-Brown et al., 2018*) and the Dipteran (fly) *Protophormia terraenovae* (1614 aa) (*Hase et al., 2017*), a 4.7-fold difference. There is a broad tendency for the size of the protein to be relatively stable within an order (*Supplementary file 1*). Thus, as noted previously (*Maier et al., 2008*), the Hymenoptera generally have a small Hairless (of the order of 400 aa; see *Figure 1A*), while the Diptera typically have a much larger version (of the order of 1000 aa or more). Notable exceptions to this pattern of uniformity are aphids, where Hairless is typically ~900 aa compared to ~400 aa in other Hemiptera, and chalcid wasps, where the protein is over 500 aa instead of the Hymenoptera-typical ~400 aa noted above (*Supplementary file 1*). Smaller Hairless proteins typically retain all five conserved motifs/domains characteristic of this factor (*Maier et al., 2008*), while the regions that flank and lie between these sequences are reduced in size (*Figure 1A*; *Supplementary file 2*).

### A known CtBP-binding motif is present in the non-conserved N-terminal region of Su(H) in a wide variety of protostomes

The apparent confinement of the Hairless co-repressor adaptor protein to the Pancrustacea raises the question of the mechanism(s) by which Su(H) in other protostomes might recruit co-repressor proteins to mediate its repressor function. Of course, other protostomes need not utilize the Gro and CtBP co-repressors for this purpose; different co-repressors might substitute. Nevertheless, we first sought to identify known binding motifs for Gro and CtBP in Su(H) from arthropods lacking Hairless. As shown in *Table 1*, we found a canonical CtBP recruitment motif of the form PφDφS (where φ = I, L, M, or V) in predicted Su(H) proteins from a variety of Myriapods and Chelicerates, including the centipede *Strigamia maritima*, the tick *Ixodes scapularis*, the spider *Parasteatoda tepidariorum*, the horseshoe crab *Limulus polyphemus*, and the scorpion *Centruroides sculpturatus*. These motifs are all located in the non-conserved N-terminal region of Su(H) (*Supplementary file 3*).

Extending this sequence analysis to other protostome phyla led to the finding that a similar PφDφS motif occurs in the N-terminal region of Su(H) from a large number of molluscs and annelids, as well as from multiple Nemertea, Brachiopoda, Phoronida, and monogonont rotifers, and also from some flatworms (*Table 1*). It is notable, by contrast, that we do not find CtBP-binding motifs present in Su(H) from nematodes. Nevertheless, given the broad phylogenetic distribution of the

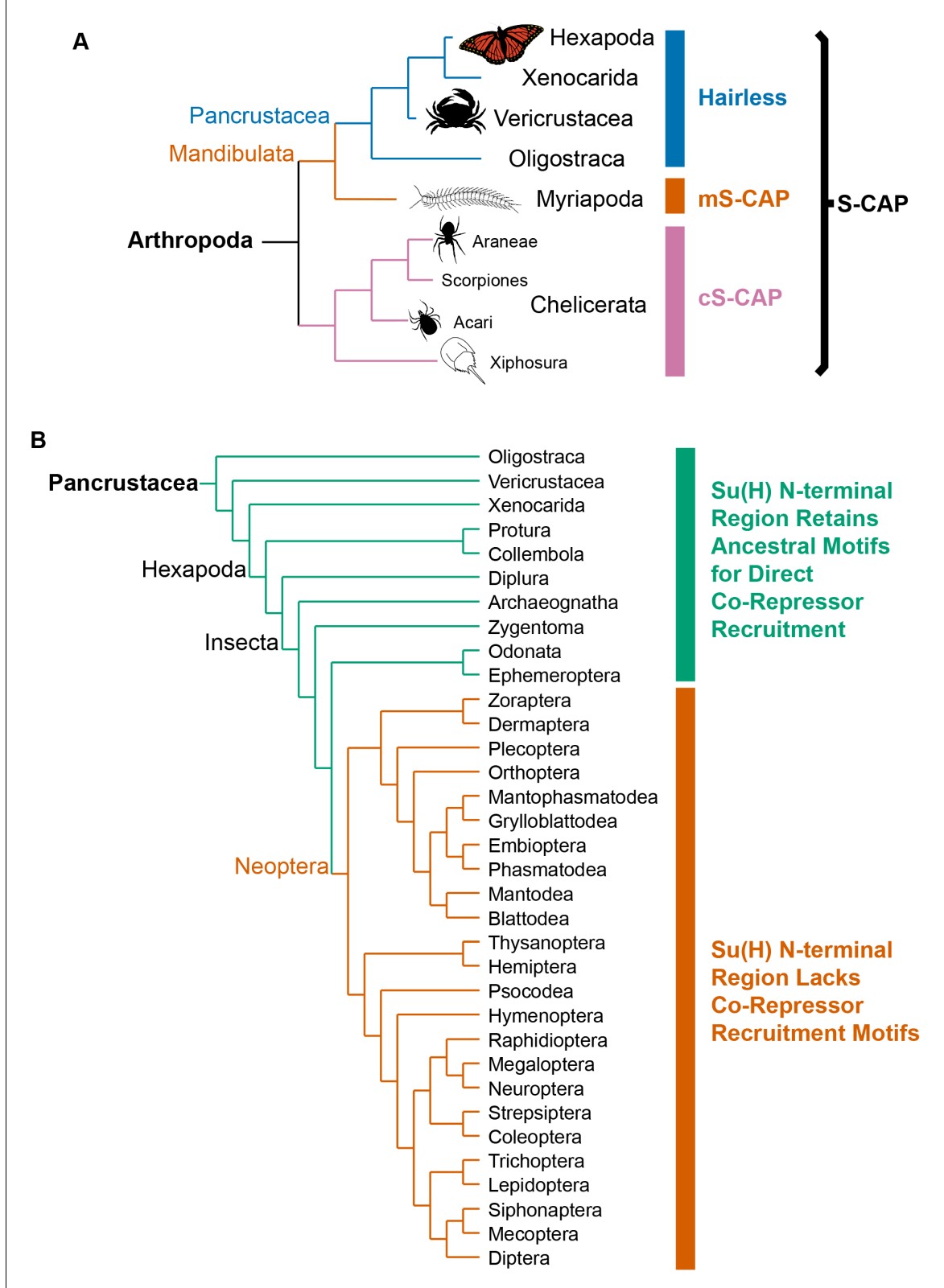

**Figure 2.** Phylogenetic distribution of Hairless and related S-CAP proteins. (**A**) Based on extensive BLAST searches of available genome and transcriptome assemblies, orthologs of the canonical *Hairless* gene are found only in the Pancrustacea (blue bar), while orthologs of a gene that encodes the related S-CAP protein are found in the Myriapods (mS-CAP, red bar) and Chelicerates (cS-CAP, pink bar). We suggest S-CAP as a suitable umbrella nomenclature for this gene family (black bracket). Tree adapted from Figure 2 of *Regier et al. (2010)*. (**B**) Consistent with the presence of
*Figure 2 continued on next page*

*Figure 2 continued*

Hairless as an adaptor protein, Su(H) in most insect orders (the Neoptera clade) has lost the ancestral short linear motifs that mediate direct recruitment of the CtBP and Gro co-repressor proteins (red bar). However, in the Crustacea, Collembola, Diplura, and a subset of Insecta, the ancestral recruitment motifs have been retained in Su(H), despite the presence of Hairless (see *Table 1* and *Supplementary file 3*). Tree adapted from *Misof et al. (2014)* and *Kjer et al. (2016)*.

DOI: https://doi.org/10.7554/eLife.48115.004

PφDφS motif in Su(H) from both Ecdysozoa and Lophotrochozoa, our observations strongly suggest that direct recruitment of CtBP by Su(H) is ancestral in the protostomes.

To verify that the shared PφDφS motif in protostome Su(H) proteins can indeed mediate direct recruitment of CtBP, we carried out an in vitro pulldown assay using GST-tagged *Drosophila* CtBP (bound to Glutathione Sepharose beads) and a His-tagged fragment of *Strigamia maritima* Su(H) (*Figure 3A*). We found that the two proteins do interact directly and robustly, in a manner that is dependent on the integrity of the PVDLS motif in *Strigamia* Su(H).

## A novel conserved motif in protostome Su(H) binds the Gro co-repressor

In addition to a PφDφS CtBP-binding motif, we have found that Su(H) from a wide variety of proto-stomes includes a novel motif similar to GSLTPPDKV (*Table 1*). Where present, this sequence typi-cally lies a short (but variable) distance C-terminal to the PφDφS motif, also within the non-conserved N-terminal region of the protein (*Supplementary file 3*). The GSLTPPDKV motif is particularly preva-lent in Su(H) from the Trochozoa, which includes annelids, sipunculans, molluscs, nemerteans, bra-chiopods, and phoronids (*Kocot et al., 2017*). Among the Ecdysozoa, it appears consistently in Su (H) from Crustacea and Myriapoda, and in small subsets of both Hexapoda (Ephemeroptera, Odo-nata, Zygentoma, Archaeognatha, Diplura, and Collembola) and Chelicerata [harvestmen (Opiliones) and Scorpiones]. The motif is absent from Su(H) in all other insect orders, and we have not found it so far in Su(H) from nematodes, flatworms, rotifers, or tardigrades; it is, however, found in the ony-chophoran *Euperipatoides kanangrensis* (*Table 1*). Perhaps surprisingly, the motif is present in Su(H) from the acorn worms *Saccoglossus kowalevskii* and *Ptychodera flava* (*Simakov et al., 2015*), which are hemichordates (deuterostomes).

Using an in vitro pulldown assay, we tested the possibility that the GSLTPPDKV motif mediates binding of the Gro co-repressor (*Figure 3B*). Indeed, we find that GST-tagged Gro protein interacts strongly with a His-tagged protein bearing this motif at its C-terminus, and that this binding is abol-ished when the motif is replaced by alanine residues. We conclude that Su(H) from a broad range of protostomes is capable of directly recruiting both CtBP and Gro (*Table 1*), and that this capacity is hence very likely to be ancestral in this clade.

## Retention of the hybrid state: Species that have both Hairless and the co-repressor-binding motifs in Su(H)

The evolutionary emergence of Hairless as an adaptor protein capable of mediating the indirect recruitment of both Gro and CtBP to Su(H) might be expected to relieve a selective pressure to retain the ancestral Gro- and CtBP-binding motifs in Su(H) itself. And indeed, we find that Su(H) from multiple insect orders comprising the Neoptera lacks both of these sequences (*Figure 2B*). Strikingly, however, we have observed that Crustacea and a small group of Hexapoda retain both traits (*Figure 2B*). Thus, multiple representatives of the Branchiopoda, Malacostraca, and Copepoda, along with Ephemeroptera, Odonata, Zygentoma, Archaeognatha, Diplura, and Collembola, have both a canonical Hairless protein (including its Gro- and CtBP-binding motifs) and Gro- and CtBP-binding motifs within Su(H). These clades, then, appear to have retained a 'hybrid intermediate' state (*Baker et al., 2012*) characterized by the presence of both co-repressor recruitment mechanisms.

## Myriapods and Chelicerates encode a protein with similarity to Hairless

While canonical Hairless proteins are confined to the Pancrustacea, we have discovered that the genomes of Myriapods and Chelicerates nevertheless encode a protein with intriguing similarities to

**Table 1.** Co-repressor recruitment motifs in protostome Su(H) proteins.

| Species | CtBP motif | Gro motif | Source |
|---|---|---|---|
| *Ecdyonurus insignis* | YPDNH**PVDLS**SPRPH | APMIP**GSLTPPDKMNGE**HPHHG | GCCL01029953.1 (*Simon et al., 2018*) |
| *Calopteryx splendens* | YTDNH**PVDLS**SPRPP | HHMIP**GSLTPPDKMNGE**HPAMH | LYUA01002621.1 (*Ioannidis et al., 2017*) |
| *Atelura formicaria* | YPDNH**PVDLS**SPRPQ | PHMIP**GSLTPPDKMNGE**HPHHS | GAYJ02050375.1 (*Misof et al., 2014*) |
| *Machilis hrabei* | YPDNH**PVDLS**SPRPH | PHMLP**GSLTPPDKMNGE**HPHHG | Scaffold 1 (*i5K Consortium, 2013*) |
| *Catajapyx aquilonaris* | STANN**PVDLS**SPRGS | APMIP**GSLTPPDKVNGE**HHSHH | JYFJ02000853.1 (*i5K Consortium, 2013*) |
| *Holacanthella duospinosa* | VPNSN**PVDLS**NPSPS | SNFVP**GSLSPPERMNGN**DPSLL | NIPM01000059.1 (*Wu et al., 2017*) |
| *Pollicipes pollicipes* | YPDNH**PVDLS**SPRPE | GPLIA**GSLTPPDKLGAE**LGLHA | GGJN01104381.1 (unpublished) |
| *Hyalella azteca* | SLGHR**PVDLS**QAPSP | AAMLA**GSLTPPDKLNSD**PQQQQ | NW_017238139.1 (*i5K Consortium, 2013*) |
| *Eurytemora affinis* | SETSA**PVDLS**APRPN | YGMLP**GSLTPPDKLNGD**HCSPG | NW_019396104.1 (*i5K Consortium, 2013*) |
| *Triops cancriformis* | HPEAR**PVDLS**SSRLL | YHSSS**LTLTPPDKVNVD**GSNSQ | BAYF01001879.1 (*Ikeda et al., 2015*) |
| *Argulus siamensis* | YPENN**PVDLS**NSRTG | SPMIP**GSLTPPDKMNGE**HHPGH | JW959185.1 (*Sahoo et al., 2013*) |
| *Strigamia maritima* | FADNH**PVDLS**NSHRG | SHMIA**GSLTPPDKVNGE**HGHQL | JH430541.1 (*Chipman et al., 2014*) |
| *Sigmoria latior munda* | TNENH**PVDLS**SSHRS | SHMIP**GSLTPPDKGNAE**HSHSH | (*Rodriguez et al., 2018*) |
| *Metaseiulus occidentalis* | GADRK**PLDMS**AAHRS | | NW_003805473.1 (*Hoy et al., 2016*) |
| *Ixodes scapularis* | QAAGA**PVDMS**SHPAR | | NW_002722632.1 (*Gulia-Nuss et al., 2016*) |
| *Parasteatoda tepidariorum* 1 | VIDSH**PVDLS**SPKPS | | NW_018383625.1 (*Schwager et al., 2017*) |
| *Parasteatoda tepidariorum* 2 | RYEGR**PVDLS**SPRPN | | NW_018370942.1 (*Schwager et al., 2017*) |
| *Limulus polyphemus* 1 | PYDGH**PVDLS**NQRPD | | NW_013671976.1 (*Battelle et al., 2016*) |
| *Limulus polyphemus* 2 | TYESH**PVDLS**NQRPD | | NW_013676581.1 (*Battelle et al., 2016*) |
| *Centuroides sculpturatus* | GYESS**PVDLS**SHRSV | MQLIS**GSMTSHDKVNGD**QHSLG | NW_019384406.1 (*Schwager et al., 2017*) |
| *Euperipatoides kanangrensis* | NSYDN**PVDLS**SHRSS | QQILP**GSLGPSDKVNGD**LVSLA | LN881712.1 (unpublished) |
| *Naineris dendritica* | DPNGH**PVDLS**HSRHI | PHMIH**GSLTPPDRVNGE**PGSGL | (*Andrade et al., 2015*) |
| *Platynereis dumerilii* | MASEN**PVDLS**SRHVG | GNHFP**GTLTPPDKLNGD**HNAHH | KP293861.1 (*Gazave et al., 2017*) |
| *Nephasoma pellucidum* | AGYET**PVDLS**SPRPC | SHLIP**GSLTPPDKINGE**GITTS | (*Lemer et al., 2015*) |
| *Owenia sp.* | QPYEN**PVDLS**RRHIK | AHLIP**GSLTPPDKINGD**MVTMA | (*Andrade et al., 2015*) |
| *Octopus bimaculoides* | NGFDN**PMDLS**NGKVV | HLMPA**GSLTPPDKISGD**SISMA | NW_014678436.1 (*Albertin et al., 2015*) |
| *Crassostrea gigas* | GGYEN**PMDLS**SNKPG | SHIVA**GSLTPPEKINGD**PGAMA | NW_011936122.1 (*Zhang et al., 2012*) |
| *Lottia gigantea* | AGVEN**PVDLS**NGRIS | SHLFT**GSLTPPEKPNGD**LVPMS | NW_008708401.1 (*Simakov et al., 2013*) |
| *Notospermus geniculatus* | VQYDN**PIDLS**NRLEG | NHMIP**GSLTPPDKVNGD**MVPLP | GFRY01035878.1 (*Luo et al., 2018*) |
| *Malacobdella grossa* | LHYDN**PLDLT**NRLDE | GSGIA**GSMTPPDGGKGN**DLDLQ | (*Whelan et al., 2014*) |
| *Lingula anatina* | GGYEN**PMDLS**RRTEM | AHMIP**GNLTPPDKVNGE**MVPMA | GDJY01029776.1 (*Luo et al., 2015*) |
| *Phoronis australis* | QHDNR**PMDLS**SRGQH | SHLIA**GSLTPPDKVNGD**VVSMA | GFSC01078935.1 (*Luo et al., 2018*) |
| *Procotyla fluviatilis* | ETLFE**PLDLR**SPIGV | | GAKZ01044347.1 (unpublished) |
| *Brachionus koreanus* | AKDET**PIDLS**SKKSK | | GBXV02009219.1 (*Lee et al., 2015*) |
| *Xenoturbella bocki* | KRYSA**PLNLT**VHDKC | DVRVL**GRLTPPDKQHVN**NDVGA | (*Brauchle et al., 2018*) |

Shown are alignments of short linear amino acid motifs (bold) in the N-terminal region of Su(H) proteins that mediate direct recruitment of the co-repressors CtBP and Gro. Column at right shows the source of the corresponding sequence data, with accession numbers and publication citations indicated.

DOI: https://doi.org/10.7554/eLife.48115.005

Hairless. Most notable is the presence of a motif that strongly resembles the 'Su(H)-binding domain' (SBD) of Hairless, which mediates its high-affinity direct interaction with Su(H) (*Figure 1*; *Figure 4A*). We will refer to these proteins as 'S-CAPs'; the basis for this designation will be made clear in forthcoming figures. We note that the occurrence of this protein in the centipede *Strigamia maritima* has also recently been reported by *Maier (2019)*. In the Pancrustacea, the N-terminal and C-terminal halves of the Hairless SBD are encoded by separate exons (*Figure 4B*). Strikingly, the related motif in Myriapod and Chelicerate S-CAPs is likewise encoded by separate exons, with exactly the same

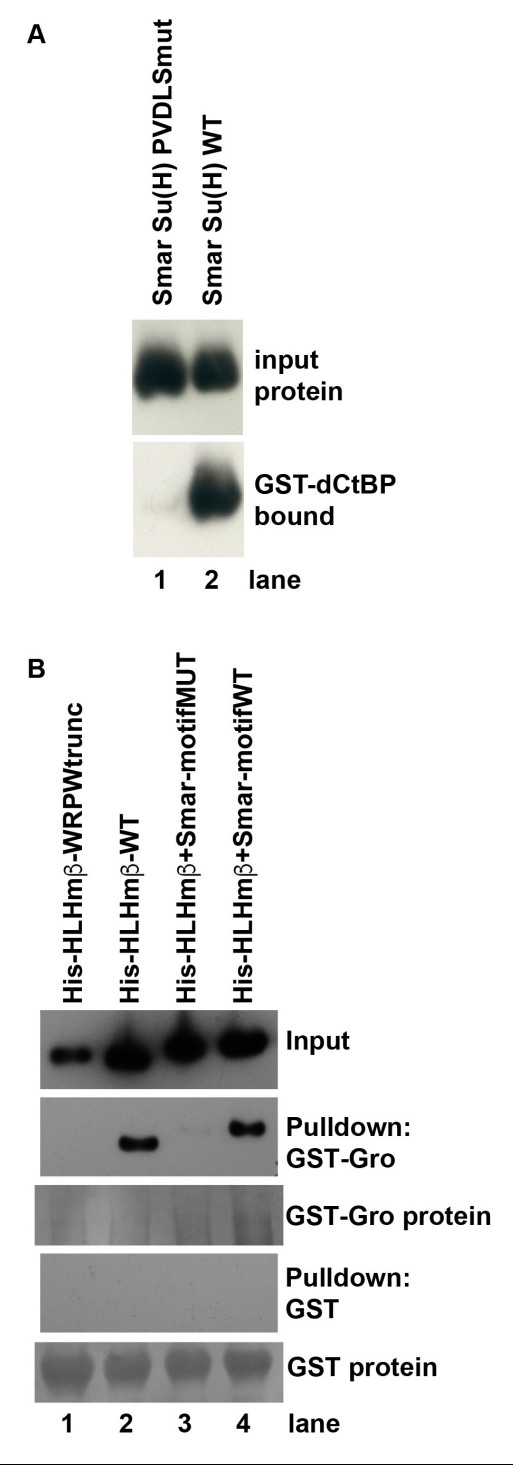

**Figure 3.** Direct binding of co-repressor proteins by Su(H) from the centipede *Strigamia maritima*. (**A**) The PVDLS motif in the N-terminal region of Su(H) from the centipede *Strigamia maritima* directly binds *Drosophila* CtBP. A His-tagged 116-aa segment of the *Strigamia* Su(H) protein, bearing a PVDLS recruitment motif for CtBP, binds strongly to GST-dCtBP (WT, lane 2). Mutation of the motif to alanines (AAAAA) abolishes

*Figure 3 continued on next page*

splice junction as in Hairless (*Figure 4B*). We believe that this is highly unlikely to be coincidental, and is instead strongly suggestive of an evolutionary relationship between Hairless and S-CAPs.

A recent structural analysis of the Su(H)-Hairless protein complex identified several residues in the Hairless SBD that are involved in binding to the C-terminal domain (CTD) of Su(H) (*Yuan et al., 2016*) (*Figure 4A*). These include four hydrophobic amino acids in the main body of the SBD (L235, F237, L245, and L247; these are highlighted in red in *Figure 4A*). Note that the Myriapod and Chelicerate S-CAP motifs share these same residues. In addition, a tryptophan (W258) C-terminal to the main body of the Hairless SBD also participates in binding to Su(H) (*Figure 4A*). Myriapod and Chelicerate S-CAPs all include a tryptophan residue at a similar position C-terminal to the main SBD-like domain (*Figure 4A*). Moreover, this particular W residue in both Hairless and the S-CAPs is followed by a hydrophobic residue, typically V or I. These sequence features, we suggest, is further strong evidence of a common ancestry for the respective segments of Hairless and S-CAPs.

A third structural similarity between Hairless and S-CAPs is the presence in the latter of one or more short linear motifs capable of binding the CtBP co-repressor (*Figure 5A*). These motifs typically reside in the C-terminal half of the S-CAPs, superficially resembling the C-terminal location of Hairless's CtBP recruitment motif.

A table listing representative examples of Myriapod and Chelicerate S-CAPs is provided as *Supplementary file 4*, and an annotated FASTA file of their amino acid sequences is included as *Supplementary file 5*. It is important to note that we have not found non-Hairless S-CAPs in the Pancrustacea.

## Spider S-CAP binds to *Drosophila* Su(H)

Given the clear sequence similarity between the Hairless SBD and the SBD-like motif in Myriapod and Chelicerate S-CAPs, we investigated whether the latter motif is likewise capable of mediating direct binding to Su(H). As noted above, the Hairless SBD interacts specifically with the CTD of Su(H). Since this domain in Su(H) is very highly conserved throughout the Bilateria and Cnidaria, we thought it reasonable to utilize *Drosophila* Su(H) for this binding assay. As shown in *Figure 4C*, we find that a 200-amino-acid segment of S-CAP from the spider *Parasteatoda tepidariorum* binds directly to *Drosophila* Su(H) in vitro. This

*Figure 3 continued*

this interaction (PVDLSmut, lane 1). The results shown in this panel have been replicated in eight additional experiments, utilizing three independent isolations of GST-CtBP protein from bacterial cultures. (B) The conserved GSLTPPDKV motif in the N-terminal region of *Strigamia* Su(H) directly binds *Drosophila* Gro. His-tagged E(spl)mβ-HLH protein, which bears a C-terminal WRPW motif that recruits Gro, is used as a binding control. Wild-type (WT) HLHmβ binds GST-Gro (lane 2), while a truncated version of the protein lacking the WRPW motif (lane 1) fails to bind. A synthetic version of HLHmβ in which the WRPW motif has been replaced by the wild-type GSLTPPDKV motif also binds GST-Gro efficiently (lane 4), while a mutant version in which GSLTPPDKV is replaced by alanines (AAAAAAAAA) shows extremely weak binding (lane 3). No binding of any of the His-tagged proteins to GST alone is observed, even with substantially greater amounts of GST compared to GST-Gro. The results shown in this panel have been replicated in seven additional experiments, utilizing five independent isolations of GST-Gro protein from bacterial cultures.

DOI: https://doi.org/10.7554/eLife.48115.006

interaction depends strictly on the integrity of the five residues that in Hairless have been shown to contact the Su(H) CTD (highlighted in red in *Figure 4A*).

Given the presence of one or more CtBP recruitment motifs in the Myriapod and Chelicerate S-CAP proteins (*Figure 5A*), along with the ability of their SBD-like domains to bind Su(H) (*Figure 4C*), we have designated these as 'Su(H)-Co-repressor Adaptor Proteins' (S-CAPs).

## Chelicerate S-CAP proteins are related to Metastasis-associated (MTA) proteins

In addition to their similarities to Hairless, the S-CAP proteins of Chelicerates include two regions with strong sequence homology to the Metastasis-associated (MTA) protein family, which is highly conserved among Metazoa. The MTA proteins play an important role in transcriptional regulation via their function as core components of the nucleosome remodeling and deacetylase (NuRD) complex (*Allen et al., 2013*; *Burgold et al., 2019*). The N-terminal half of MTAs includes four well-defined functional domains: BAH (Bromo-Adjacent Homology), ELM2 (Egl-27 and MTA1 homology), SANT (Swi3, Ada2, N-CoR, and TFIIIB), and GATA-like zinc finger (*Millard et al., 2014*) (*Figure 5B*). Of these, the ELM2 and SANT domains are retained at the N-terminal end of Chelicerate S-CAPs (*Figure 5B*; *Figure 5—figure supplement 1A*). This is highly likely to have functional significance, as the ELM2 and SANT domains of MTA proteins work together to recruit and activate the histone deacetylases HDAC1 and HDAC2 (*Millard et al., 2013*). Further suggesting homology between Chelicerate S-CAPs and MTAs is the observation that their shared ELM2 and SANT domains are each encoded by two exons with exactly the same splice junction (*Figure 5C*).

It is noteworthy that, despite sharing the SBD-like and CtBP recruitment motifs of Chelicerate S-CAPs, the available Myriapod S-CAP protein sequences lack the N-terminal ELM2 and SANT homologies with MTA proteins (*Figure 5B*). Consistent with this, the SBD motif in Myriapod S-CAPs lies much closer to the protein's N terminus than the SBD motif in Chelicerate S-CAPs, suggesting that simple loss of the ELM2/SANT-encoding exons might underlie this difference between the two S-CAP clades. Likewise, Hairless proteins are devoid of clear similarities to MTAs.

In addition to their SBD and ELM2/SANT domains, Chelicerate S-CAPs share a third region of homology that lies between the ELM2 and SANT sequences (*Figure 5—figure supplement 1A*). This region is absent from both Hairless and the Myriapod S-CAPs. Conversely, Myriapod S-CAPs include a segment of sequence similarity that is not found in either Hairless or Chelicerate S-CAPs (*Figure 5—figure supplement 1B*).

## Conserved microsynteny between *MTA* and *S-CAP*/*Hairless* genes

Our analysis of the genomic locations of genes encoding MTA proteins in Arthropoda, Hairless in Pancrustacea, and S-CAPs in Myriapods and Chelicerates has yielded the surprising finding that proximate or near-proximate linkage between *MTA* and *Hairless* genes or between *MTA* and *S-CAP* genes is broadly conserved among arthropods (*Figure 6*; *Supplementary file 1*; *Supplementary file 4*). Thus, in the centipede *Strigamia maritima*, the gene encoding S-CAP lies immediately upstream of that encoding MTA, in the same orientation (*Figure 6*; *Supplementary file 4*). A similar linkage relationship between *S-CAP* and *MTA* genes is seen in many arachnids, including the spiders *Nephila clavipes* (*Supplementary file 4*) and *Parasteatoda tepidariorum* (*Figure 6*; *Supplementary file 4*) and the mites *Achiptera coleoptrata* and *Sarcoptes scabiei* (*Supplementary file 4*). Likely due at

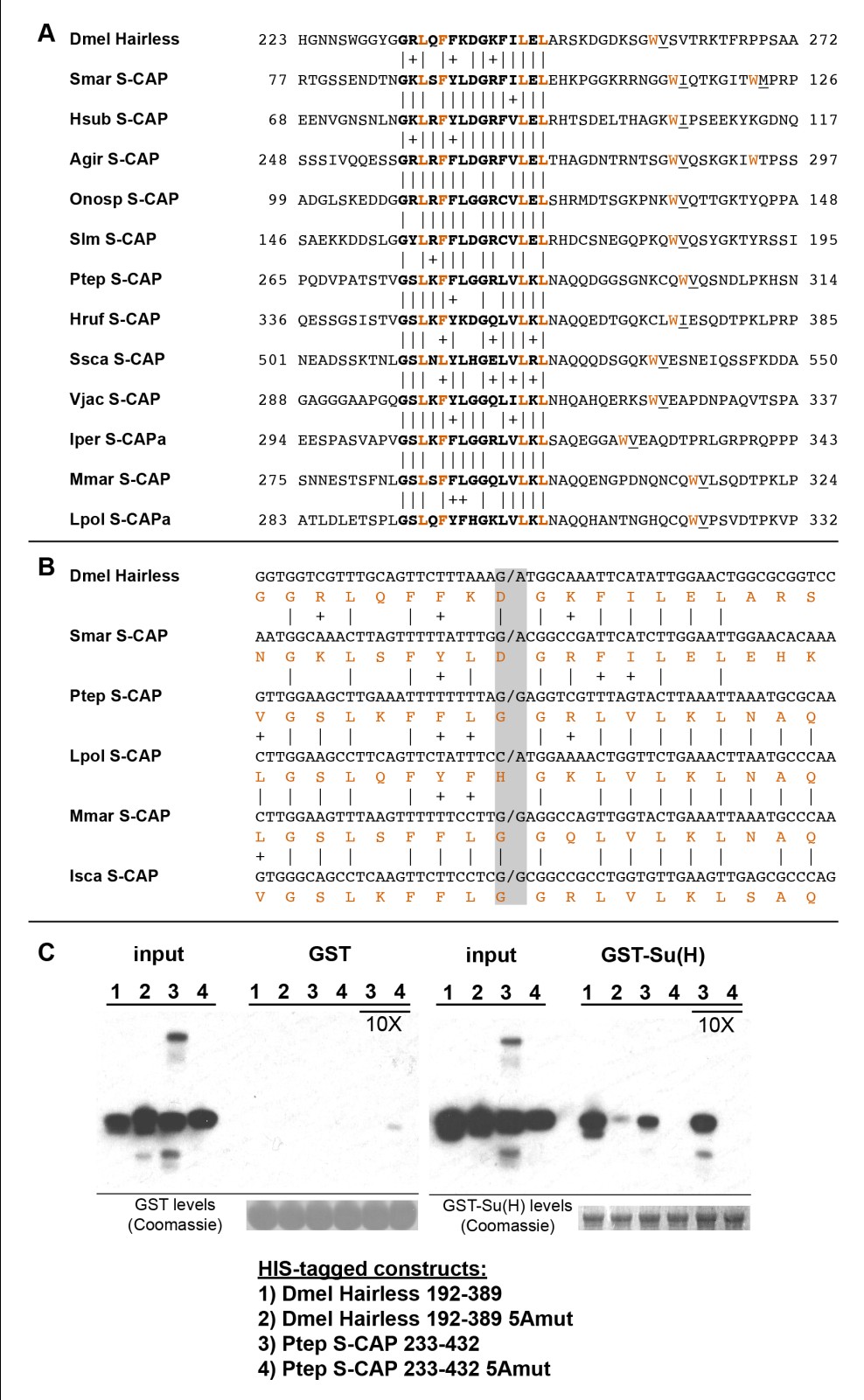

**Figure 4.** S-CAP proteins in Myriapods and Chelicerates contain a Hairless-like domain that binds Su(H). (**A**) Alignment of the Suppressor of Hairless Binding Domain (SBD) in *Drosophila melanogaster* (Dmel) Hairless with the related motif in the S-CAP proteins from a representative set of Myriapods and Chelicerates. Numbers flanking each sequence segment represent amino acid positions within the protein. The contiguous SBD motif is
*Figure 4 continued on next page*

*Figure 4 continued*

highlighted in bold. Pairwise amino acid sequence identities within the motifs are indicated by vertical lines; conservative substitutions are indicated by + signs. Amino acids in Hairless that have been shown to make direct contact with Su(H) [including the non-contiguous tryptophan (W) residue] (*Yuan et al., 2016*) are highlighted in red. Hydrophobic residues nearly always found immediately adjacent to the W are underlined. Species names are as follows: Smar (*Strigamia maritima*); Hsub (*Hydroschendyla submarina*) (*Fernández et al., 2016*); Agir (*Anopsobius giribeti*) (*Fernández et al., 2016*); Onosp (*Onomeris* sp.) (*Rodriguez et al., 2018*); Slm (*Sigmoria latior munda*) (*Rodriguez et al., 2018*); Ptep (*Parasteatoda tepidariorum*); Hruf (*Hypochthonius rufulus*) (*Bast et al., 2016*); Ssca (*Sarcoptes scabiei*); Vjac (*Varroa jacobsoni*) (*Techer et al., 2019*); Iper (*Ixodes persulcatus*); Mmar (*Mesobuthus martensii*) (*Cao et al., 2013*); Lpol (*Limulus polyphemus*). We note that *Maier (2019)* has previously described the presence of the SBD-like element in the *Strigamia maritima* sequence. (B) SBD motifs in both Hairless and S-CAP proteins (red) are encoded in two exons with the same splice junction (indicated by /; see gray highlight). Pairwise amino acid sequence identities within the motifs are indicated by vertical lines; conservative substitutions are indicated by + signs. Species names as in A, except for Isca (*Ixodes scapularis*). (C) Spider S-CAP protein binds directly to *Drosophila* Su(H) in vitro. In all panels, lanes 1–4 represent the indicated His-tagged segments of wild-type *Drosophila* (Dmel) Hairless (lane 1); Dmel Hairless bearing alanine substitutions for each of five SBD residues shown to contact Su(H) (lane 2); wild-type S-CAP from the spider *Parasteatoda tepidariorum* (Ptep) (lane 3); Ptep S-CAP bearing the same five alanine substitutions (lane 4). Input levels of these His-tagged proteins for each experiment are shown in the respective 'input' panels. Remaining two panels show the results of pulldown assays using Sepharose beads bearing only GST (left side) or GST-Su(H) (right side). Left: No binding of the His-tagged proteins to GST alone is observed. Right: Wild-type Dmel Hairless binds efficiently to GST-Su(H) (lane 1); this interaction is severely reduced by the introduction of the five alanine substitutions (lane 2). Wild-type Ptep S-CAP likewise binds to GST-Su(H) (lane 3), while no binding is observed with the alanine-substitution mutant (lane 4); the same result is obtained even when the amount of input Ptep S-CAPs (wild-type and mutant) is increased by a factor of 10 (lanes 3 and 4, 10X). Amounts of GST and GST-Su(H) on the beads are shown in the Coomassie stains below the corresponding pulldown lanes. The results shown in this panel have been replicated in two additional experiments, including one utilizing new isolations of GST-Su(H) and His-tagged proteins.

DOI: https://doi.org/10.7554/eLife.48115.007

least in part to its history of whole-genome duplication (*Nossa et al., 2014*; *Kenny et al., 2016*), the horseshoe crab *Limulus polyphemus* (representing the Merostomata/Xiphosura) has three paralogous copies of this same *S-CAP-MTA* linkage pairing (*Supplementary file 4*). Some exceptions to this pattern do exist. In the genomes of the mites *Metaseiulus occidentalis* (*Supplementary file 4*) and *Varroa destructor* (*Techer et al., 2019*), for example, the genes encoding S-CAP and MTA are far separated from each other.

Close, typically adjacent, linkage between *Hairless* and *MTA* genes is likewise widely observed in the genomes of Pancrustacea. Among the Hexapoda, this pattern can be found in many different orders (*Supplementary file 1*), including Diptera, Lepidoptera, Coleoptera (*Figure 6*), Hymenoptera (*Figure 6*), Psocodea, Hemiptera (*Figure 6*), Thysanoptera, Blattodea, Orthoptera, Odonata, and Collembola. Among the Vericrustacea, adjacent linkage of *Hairless* and *MTA* is seen in the shrimp *Triops cancriformis* (Notostraca) (*Supplementary file 1*). Nevertheless, exceptions are readily found, even within the same orders as above (*Supplementary file 1*). Examples include *Drosophila melanogaster*, *Ceratitis capitata*, and *Lucilia cuprina* (Diptera; *Supplementary file 1*), *Bicyclus anynana* (Lepidoptera), *Anoplophora glabripennis*, *Dendroctonus ponderosae*, and *Nicrophorus vespilloides* (Coleoptera), and *Cimex lectularius* (Hemiptera; *Supplementary file 1*).

Interestingly, in some instances *Hairless/MTA* microsynteny is preserved, but the genes' relative orientation is different (*Figure 6*; *Supplementary file 1*). Thus, in the aphids — in contrast to other Hemiptera — *MTA* lies downstream of *Hairless*, but in the opposite orientation (*Figure 6*). In the beetle *Harmonia axyridis* (Coleoptera), *MTA* lies upstream of *Hairless* (*Figure 6*).

Despite the multiple instances in which it has been lost, we believe that the most parsimonious interpretation of our analysis is that close linkage between *MTA* and *S-CAP/Hairless* genes is ancestral in the respective taxa (Myriapods/Chelicerates and Pancrustacea). We leave for the Discussion our proposed interpretation of the evolutionary significance of this adjacency.

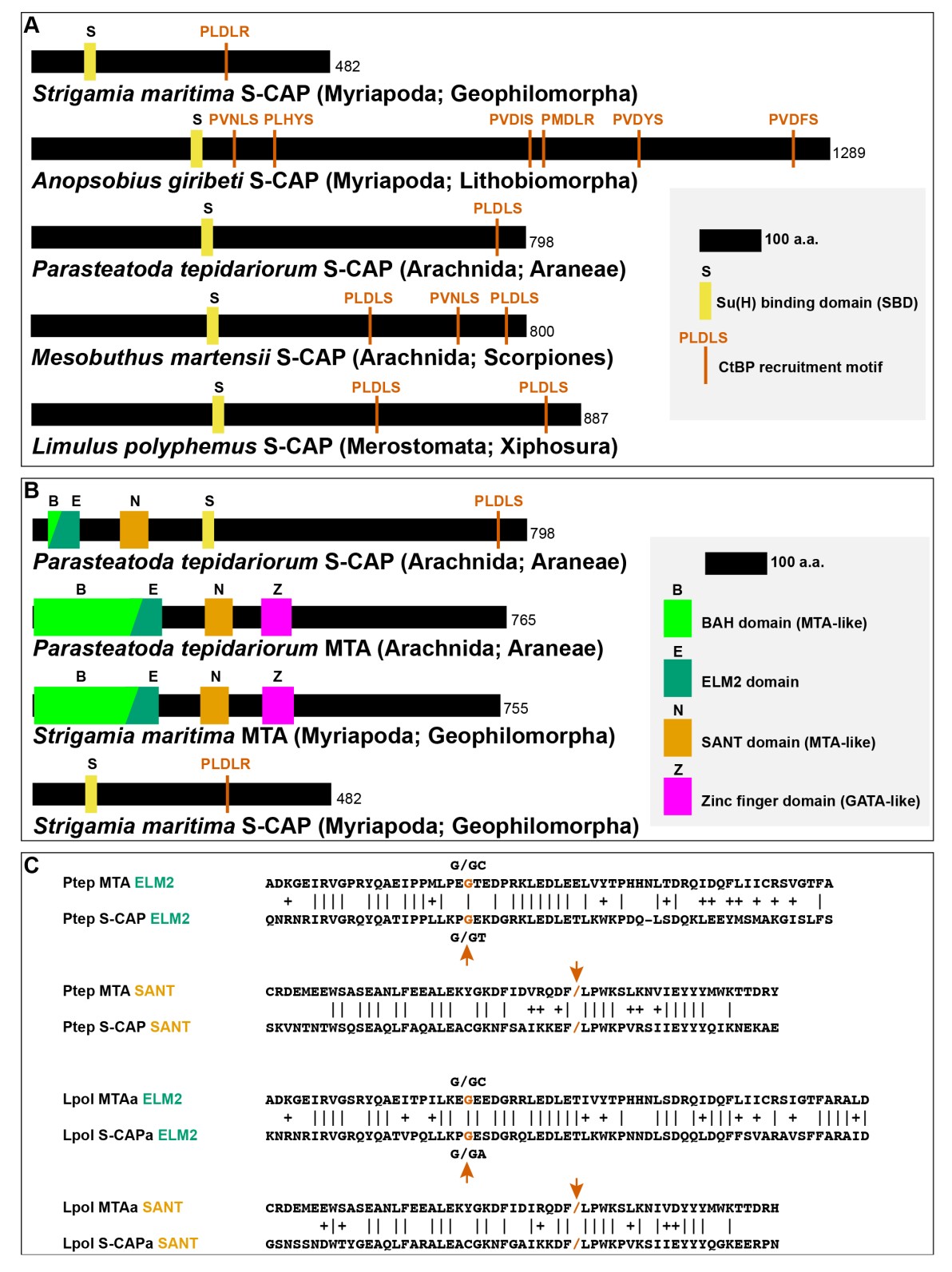

**Figure 5.** Sequence characteristics of S-CAP proteins in Myriapods and Chelicerates. (**A**) Diagrams of representative examples of Myriapod and Chelicerate S-CAP proteins, denoting locations of SBD motifs and CtBP recruitment motifs. Scale and protein sizes are indicated. (**B**) Chelicerate, but not Myriapod, S-CAP proteins share N-terminal ELM2 and SANT domains with an MTA zinc-finger protein from the same species. Scale and protein sizes are indicated. (**C**) Shared ELM2 and SANT domains in Chelicerate MTA and S-CAP proteins are encoded in two exons with the same splice

*Figure 5 continued on next page*

*Figure 5 continued*

junction (indicated by /; red arrows). Pairwise amino acid sequence identities within the motifs are indicated by vertical lines; conservative substitutions are indicated by + signs. Species names as in *Figure 4A*.

DOI: https://doi.org/10.7554/eLife.48115.008

The following figure supplement is available for figure 5:

**Figure supplement 1.** Alignments of sequence regions shared by representative S-CAP proteins from (**A**) Chelicerates and (**B**) Myriapods.

DOI: https://doi.org/10.7554/eLife.48115.009

## Discussion

### The evolution of Hairless represents a shift from the ancestral and dominant paradigm of direct co-repressor recruitment by Su(H)

Our analysis of sequences from a broad range of protostomes strongly suggests that direct recruitment of the CtBP and Gro co-repressors by Su(H) is ancestral in this clade. This is consonant with the fact that direct co-repressor recruitment by DNA-binding repressor proteins in general is a dominant paradigm among Metazoa. This evokes the intriguing question of what might have led to the loss of direct recruitment by Su(H) in the Neoptera (see *Figure 1B*) and its replacement by Hairless-mediated indirect recruitment? Does Hairless provide some advantageous functional capacity? Note that this is not intended to suggest that Hairless must be an evolutionary adaptation per se (*Lynch, 2007*);

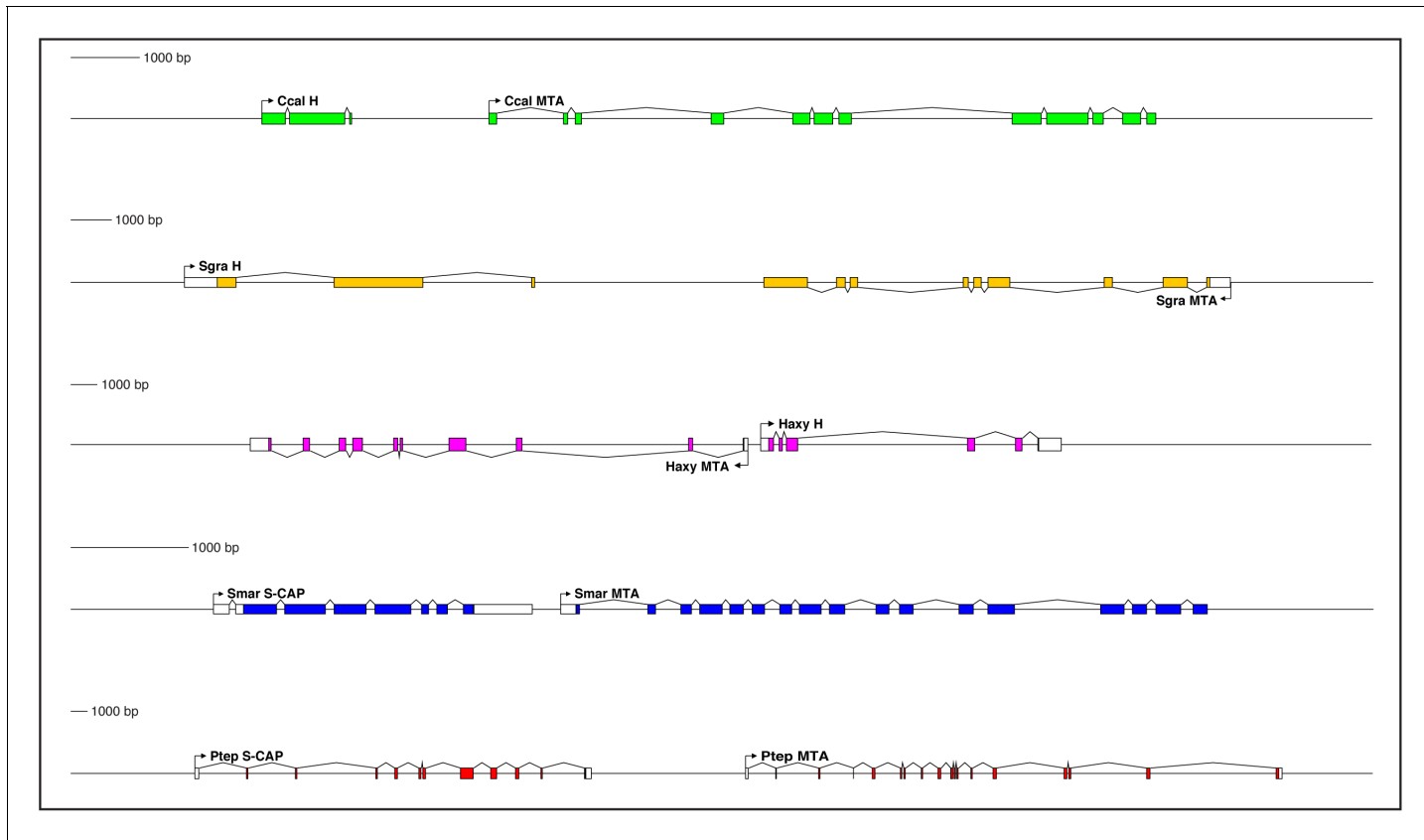

**Figure 6.** Genes encoding both Hairless and S-CAP proteins are frequently located immediately adjacent to an *MTA* gene. Separate scale for each diagram is shown at the left. Three examples are shown for *Hairless*: the carpenter bee *Ceratina calcarata* (Ccal), the wheat aphid *Schizaphis graminum* (Sgra) (QEWZ01001380.1), and the lady beetle *Harmonia axyridis* (Haxy). Note that microsynteny is often preserved even when gene locations and relative orientations are changed. One example each is shown for *S-CAP* in Myriapods [the centipede *Strigamia maritima* (Smar)] and Chelicerates [the house spider *Parasteatoda tepidariorum* (Ptep)]. See also *Supplementary file 1* and *Supplementary file 4*.

DOI: https://doi.org/10.7554/eLife.48115.010

rather, we are asking: What capability might it have conferred that would lead to its retention and the subsequent loss of the recruitment motifs in Su(H)?

One appealing (but of course speculative) possibility is that Hairless may have permitted Su(H) for the first time to recruit both CtBP and Gro simultaneously to the same target genes. As we have noted, the apparently ancestral PφDφS and GSLTPPDKV motifs in protostome Su(H) typically lie quite close to each other in the protein's linear sequence (*Supplementary file 3*). CtBP (~400 aa) and Gro (~700 aa) are both large proteins that engage in oligomerization as part of their functional mechanism (*Song et al., 2004*; *Bhambhani et al., 2011*). It is very unlikely that both could bind stably to DNA-bound Su(H) at the same time. In contrast, the Gro and CtBP recruitment motifs in Hairless are far apart in the linear sequence (*Figure 1A*) and are separated by a region predicted to be largely disordered (*Figure 1—figure supplement 1*). We suggest that this might be compatible with simultaneous recruitment of the two co-repressors.

Whatever may have been the selective forces that led to the loss of direct co-repressor recruitment by Su(H) in the Neoptera and its replacement by Hairless-mediated indirect recruitment, Hairless is a notable evolutionary novelty for having permitted the unusual abandonment of an ancestral and highly conserved paradigm. We suggest that this represents a striking example of 'developmental system drift' (*True and Haag, 2001*), in which a common output (widespread 'default repression' of Notch pathway target genes) is achieved via distinct molecular mechanisms in different species.

## A possible evolutionary pathway for the appearance of Hairless

We have described here several findings that we believe have important implications for an attempt to reconstruct the history of Hairless as an evolutionary novelty. First, we observe that Hairless is apparently confined to the Pancrustacea, wherein it is widely distributed among diverse taxa (*Figure 2A*; *Supplementary file 1*). Second, we have discovered in the sister groups Myriapoda and Chelicerata a protein (S-CAP) with clear sequence homology to the Su(H)-binding domain (SBD) of Hairless (*Figure 4A*). Significantly, in both Hairless and the S-CAPs these motifs are encoded by contributions from two exons, with the associated splice junction in precisely the same location (*Figure 4B*; *Supplementary file 4*). Third, we find that S-CAPs in the Chelicerata include in their N-terminal region strong homology to the ELM2 and SANT domains of MTAs, which themselves are highly conserved among Metazoa, and therefore would have been present in the arthropod common ancestor (*Figure 5B,C*). Finally, our analysis indicates that close, usually adjacent, linkage of *Hairless* and *MTA* genes (in the Pancrustacea) and between *S-CAP* and *MTA* genes (in the Myriapoda and Chelicerata) is widespread (*Figure 6*; *Supplementary file 1*; *Supplementary file 4*), and hence very likely to be ancestral, in these taxa.

While any attempt to infer the sequence of evolutionary events that led to the appearance of Hairless is necessarily speculative, we believe that the above findings offer substantial support for the following hypothetical pathway. We propose that in a deep arthropod ancestor a tandem duplication of the *MTA* gene occurred. One copy retained the strong sequence conservation (and presumably ancestral function) of metazoan *MTA* genes, while the second copy diverged very substantially, eventually encoding a protein that had lost all but the ELM2 and SANT domains of the MTA ancestor. The extensive reconfiguration of this paralog also included the eventual acquisition of the SBD motif and the addition of one or more CtBP recruitment motifs (see *Figure 7* for some possible sources of these components). In the Myriapod lineage, even the ELM2 and SANT domains were eventually lost. In the Pancrustacea, we suggest that this same divergent *MTA* paralog evolved to become *Hairless*. Beyond the alterations described for the Myriapoda, this would have involved the acquisition of sequences encoding additional now-conserved domains and motifs, including the Gro recruitment motif (*Supplementary file 2*). This radical evolutionary transformation resulted in a protein with little or no remaining homology to its MTA ancestor, and with an entirely novel regulatory function (*Holland et al., 2017*).

In this context, it is of interest that the *Drosophila* Mi-2/Nurd complex — which includes the MTA protein — has recently been shown to engage in direct repression of multiple Notch pathway target genes, independent of both Su(H) and Hairless (*Zacharioudaki et al., 2019*). Whether this activity preceded the emergence of Hairless is unknown, but the possibility that it is in some way connected to Hairless's evolutionary history is indeed intriguing.

# Materials and methods

## Key resources table

| Reagent type (species) or resource | Designation | Source or reference | Identifiers | Additional information |
|---|---|---|---|---|
| Antibody | anti-HIS G antibody, mouse monoclonal | Invitrogen | CAT#46–1008 (now ThermoFisher CAT#R940-25), RRID:AB_2556557 | 1:5000 dilution |
| Antibody | GOAT anti-mouse HRP, polyclonal | Jackson Immuno-research | CAT#115-035-003, RRID:AB_10015289 | 1:10000 dilution |
| Recombinant DNA reagent | GST-dCtBP pGEX-5X-3 clone | *Nibu et al., 1998* | | Construct encoding GST-tagged *Drosophila* CtBP for expression in *E. coli* |
| Recombinant DNA reagent | GST-Gro pGEX-KG clone | This paper | | Construct encoding GST-tagged *Drosophila* Groucho for expression in *E. coli* |
| Recombinant DNA reagent | GST-Su(H) pGEX-KG clone | *Bailey and Posakony, 1995* | | Construct encoding GST-tagged *Drosophila* Su(H) for expression in *E. coli* |
| Recombinant DNA reagent | HIS-H192-389 WT pRSET-C clone | This paper | | HIS-tagged expression construct encoding amino acids 192–389 of *Drosophila* Hairless, synthesized by GeneWiz, Inc, and codon-optimized for expression in *E. coli* |
| Recombinant DNA reagent | HIS-H192-389 5AMUT pRSET-C clone | This paper | | HIS-tagged expression construct encoding amino acids 192–389 of *Drosophila* Hairless with five alanine substitutions, synthesized by GeneWiz, Inc, and codon-optimized for expression in *E. coli* |
| Recombinant DNA reagent | HIS-HLHmBetaSmar WT pRSET-C clone | This paper | | HIS-tagged expression construct encoding *Drosophila* HLHmBeta with the last four amino acids (WRPW) replaced with the sequence GSLTPPDKV |
| Recombinant DNA reagent | HIS-HLHmBetaSmar MUT pRSET-C clone | This paper | | HIS-tagged expression construct encoding *Drosophila* HLHmBeta with the last four amino acids (WRPW) replaced with the sequence AAAAAAAAA |
| Recombinant DNA reagent | HIS-HLHmBetaWT pRSET-C clone | This paper | | HIS-tagged expression construct encoding full-length *Drosophila* HLHmBeta |
| Recombinant DNA reagent | HIS-HLHmBetatrunc pRSET-C clone | This paper | | HIS-tagged expression construct encoding *Drosophila* HLHmBeta with the last four amino acids (WRPW) deleted |

*Continued on next page*

| Reagent type (species) or resource | Designation | Source or reference | Identifiers | Additional information |
|---|---|---|---|---|
| Recombinant DNA reagent | HIS-PtepSCAP233-432 WT pRSET-C clone | This paper | | HIS-tagged expression construct encoding amino acids 233–432 of *Parasteatoda tepidariorum* S-CAP, synthesized by GeneWiz, Inc, and codon-optimized for expression in *E. coli* |
| Recombinant DNA reagent | HIS-PtepSCAP233-432 5AMUT pRSET-C clone | This paper | | HIS-tagged expression construct encoding amino acids 233–432 of *Parasteatoda tepidariorum* S-CAP with five alanine substitutions, synthesized by GeneWiz, Inc, and codon-optimized for expression in *E. coli* |
| Recombinant DNA reagent | HIS-SmarSu(H)ex2-3 WT pRSET-C clone | This paper | | HIS-tagged expression construct containing exons 2–3 of *Strigamia maritima Su(H)*, synthesized by GeneWiz, Inc, and codon-optimized for expression in *E. coli* |
| Recombinant DNA reagent | HIS-SmarSu(H)ex2-3 mut pRSET-C clone | This paper | | HIS-tagged expression construct containing exons 2–3 of *Strigamia maritima Su(H)* with a PVDLS > AAAAA mutation, synthesized by GeneWiz, Inc, and codon-optimized for expression in *E. coli* |
| Recombinant DNA reagent | pGEX-5X-3 | Sigma (formerly Amersham; discontinued) | CAT#28-9545-53 | |
| Recombinant DNA reagent | pRSET-C | Invitrogen | CAT#V35120 | |
| Commercial assay or kit | Chem Illumination Reagents | Pierce ECL Western Blotting Substrate | CAT#32209 | |
| Resource, sequence database | NCBI | NCBI | RRID:SCR_006472 | |
| Software, algorithm | NCBI BLAST | NCBI | RRID:SCR_004870 | |
| Software, algorithm | GenePalette | *Smith et al., 2017*;*Rebeiz and Posakony, 2004*;http://www.genepalette.org | | |
| Software, algorithm | DNA Strider | *Marck, 1988*; *Douglas, 1995* | | |
| Software, algorithm | BlastStation-Local64 | TM Software, Inc | | |

## Sequence searches, analysis, and annotation

Genome and transcriptome sequences encoding Hairless, Suppressor of Hairless, S-CAP, and MTA proteins from a wide variety of species were recovered via BLAST searches, using either the online version at the NCBI website (*Boratyn et al., 2013*) or the version implemented by the BlastStation-Local64 desktop application (TM Software, Inc). Sequences were analyzed and annotated using the GenePalette (*Rebeiz and Posakony, 2004*; *Smith et al., 2017*) and DNA Strider (*Marck, 1988*; *Douglas, 1995*) desktop software tools. Analysis of predicted disordered regions in Hairless was

conducted using DISOPRED3 on the PSIPRED server (*Buchan et al., 2013*; *Jones and Cozzetto, 2015*).

## Generation of constructs for GST pulldown experiments

*Strigamia maritima* Su(H) protein constructs to test CtBP binding

A codon-optimized fragment corresponding to exons 2 and 3 from *S. maritima* Su(H) mRNA was synthesized by Genewiz, Inc, and cloned into pRSET-C using Acc65I and BamHI restriction sites. The CtBP-motif mutant was subsequently generated by overlap extension PCR using the primers HISsmarSUH-f (CGCTGGATCCGCGGCCAGTATGAC), HISsmarSUH-r (CCATGGTACCAGTTATGCGTGGTG), HISsmarSUHctbpm-f (AACCACgCCGcaGcTGcGgCTAACAGCCATCGCGGTGAAGGCGGCCAC), HISsmarSUHctbpm-r (GCTGTTAGcCgCAgCtgCGGcGTGGTTGTCGGCGAAGTGAGGGGTCAG). After sequence confirmation, this fragment was also cloned into pRSET-C using the same enzymes. Binding of these constructs to *Drosophila melanogaster* CtBP was assayed using GST alone and a GST-CtBP fusion protein (*Nibu et al., 1998*).

Constructs to test potential Gro-binding motif in *Strigamia maritima* Su(H)

A truncated version of HLHmβ (HLHmβ-WRPWtrunc) was amplified from a pRSET-HLHmβ-WT construct using the primers HISmbeta-f (cgatggatccgaATGGTTCTGGAAATGGAGATGTCCAAG) and HISmbetatrunc-r (ccatggtaccagTCACATGGGGGCCagaggtggagctggcctcgctgggcgc); a version of HLHmβ with the WRPW motif replaced with the amino acids GSLTPPDKV (HLHmβ+Smar-motifWT) was amplified from the WT construct with HISmbeta-f and mbetaSmarSuH-r (ccatggtaccagTCACACTTTATCAGGTGGAGTGAGAGAACCCATGGGGGCCagaggtggagctggcc); and a version of HLHmβ with the WRPW motif replaced with a stretch of 9 alanine residues (HLHmβ+Smar-motifMUT) was amplified using HISmbeta-f and mbetaSmarSuHmut-r (ccatggtaccagTCAggctgccgctgcggctgccgctgctgcCATGGGGGCCagaggtggagctggcc). Each construct was then subsequently cloned into pRSET-C using the restriction enzymes BamHI and Acc65I and sequence verified. Binding of these constructs to *Drosophila melanogaster* Gro was assayed using GST alone and a GST-Gro fusion protein. The latter construct was made by cloning the full-length Gro coding sequence into the pGEX-KG expression vector at the EcoRI and SalI restriction sites: <u>gtggcgaccatcctccaaaatcggatctggttccgcgtggatccccgggaatttccggtggtggtggtggaattct</u>a**ATG**...**TAA**ATCCACAAAACCATGCAGTTTTTTCATTTTGTAATAAGCTCGTATAGTTTTTATTACAACATGTTCGAAATCATGCA*cccgggctgcaggaattcgatatcaagcttatcgatacc*<u>gtcgactcgagctcaagcttaattcatcgtgactgactgacgatctg</u> (underlined = pGEX KG vector; uppercase = *gro* cDNA; bold = *gro* start and stop codons; italic = linker)

S-CAP/Hairless constructs for Su(H) interaction analysis

Codon-optimized fragments from *Drosophila melanogaster* Hairless (residues 192–389), and *Parasteatoda tepidariorum* cS-CAP (residues 233–432) as well as 5-alanine mutant substitutions (Dmel: G**GRLQFFKDGKFILEL**ARSKDGDKSG**W** - > G**GRAQAFKDGKFIAEA**ARSKDGDKSG**A**; Ptep: V**GSLKFFLGGRLVLKL**NAQQDGGSGNKCQ**W** - > V**GSAKAFLGGRLVAKA**NAQQDGGSGNKCQ**A**) were synthesized by Genewiz, Inc. Inserts were subsequently cloned into pRSET-C using the restriction enzymes BamHI and Acc65I. Binding of these constructs to *Drosophila melanogaster* Su(H) was assayed using GST alone and a GST-Su(H) fusion protein (*Bailey and Posakony, 1995*).

GST pulldowns using each of the above constructs were performed as previously described (*Fontana and Posakony, 2009*).

## Synthesized, codon-optimized sequences

>Smar Su(H)ex2-3 WT (116 aa)

CGCTGGATCCGCGGCCAGTATGACTACCCGCCGCCGTTAGCCAGCACATACAGCCGCGAGGCCGACCTGTGGAACGTGAACCTGGCCACCTACAGCAGCGCACCGACCACATGCACCGGTGCAACCCCGGCACCTAGCGTTACCGGTTTCTACGCCCAGGCCACCGGCAGCAACAGCGTTAGCCCGAGTAGCGTGAGCCTGACCACCCTGACCCCTCACTTCGCCGACAACCACCCGGTGGACCTGAGCAACAGCCATCGCGGTGAAGGCGGCCACCTGGATCTGGTGCGCTTCCAGAGCGACCGCGTGGATGCCTACAAGCACGCCAACGGCCTGAGCGTGCATATCCCGGACCACCACGCATAACTGGTACCATGG

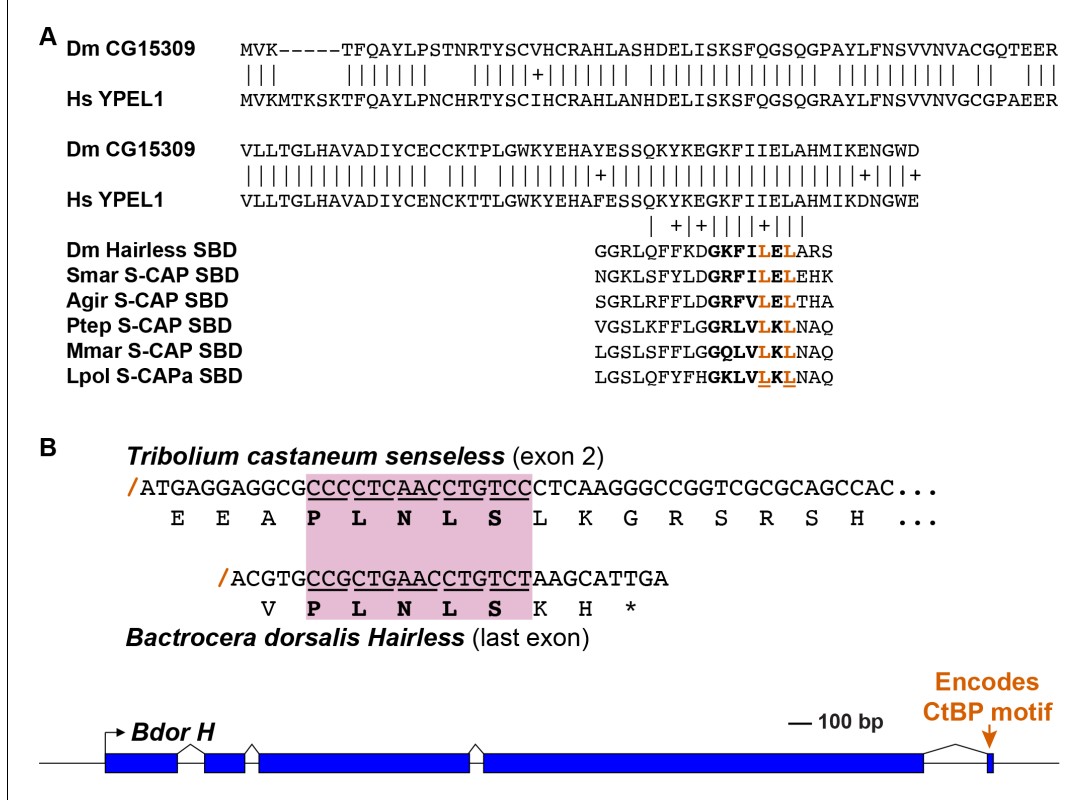

**Figure 7.** Speculative sources for key elements of the Hairless and S-CAP proteins. Note that these are intended to be only illustrative examples; other sources are of course possible. (**A**) A C-terminal segment of the highly conserved Yippee-like protein (**Roxström-Lindquist and Faye, 2001**; **Hosono et al., 2004**) is closely related to the C-terminal half of Hairless and S-CAP SBDs. Upper diagram is a sequence alignment of the entire Yippee-like proteins from *Drosophila melanogaster* (Dm) and *Homo sapiens* (Hs). Aligned below are contiguous SBD motifs from *Drosophila* Hairless and five Myriapod and Chelicerate S-CAPs; their C-terminal halves are shown in bold. Two leucine (L) residues shown to make direct contact with Su(H) (**Yuan et al., 2016**) are highlighted in red. Amino acid sequence identities are indicated by vertical lines; conservative substitutions are indicated by + signs. Other species names as in *Figure 4A*. (**B**) As shown in the gene diagram at the bottom, the CtBP recruitment motif in Hairless is encoded by a very small exon located at the extreme 3' end of the gene [example is from the Oriental fruit fly *Bactrocera dorsalis* (Bdor; JFBF01000273.1); scale indicated]. A pre-existing gene encoding a protein that utilizes the same PLNLS recruitment motif is a possible source of this exon. Example shown is a portion of the *senseless* gene from the red flour beetle *Tribolium castaneum* (**Tribolium Genome Sequencing Consortium, 2008**). Senseless directly recruits the CtBP co-repressor via the PLNLS motif (**Miller et al., 2014**). This portion of the protein is encoded in exon 2; splice junction is indicated by a red /. Aligned beneath it is the last exon of the Bdor *Hairless* gene, illustrating its splice junction in the same frame as *senseless* exon 2.
DOI: https://doi.org/10.7554/eLife.48115.011

>Smar Su(H)ex2-3 mut
CGCTGGATCCGCGGCCAGTATGACTACCCGCCGCCGTTAGCCAGCACATACAGCCGCGAGGCC-
GACCTGTGGAACGTGAACCTGGCCACCTACAGCAGCGCACCGACCACATGCACCGG
TGCAACCCCGGCACCTAGCGTTACCGGTTTCTACGCCCAGGCCACCGGCAGCAACAGCG
TTAGCCCGAGTAGCGTGAGCCTGACCACCCTGACCCCTCACTTCGCCGACAACCAC
gCCGcaGcTGcGgCTAACAGCCATCGCGGTGAAGGCGGCCACCTGGATCTGGTGCGCTTCCA-
GAGCGACCGCGTGGATGCCTACAAGCACGCCAACGGCCTGAGCGTGCATATCCCGGACCAC-
CACGCATAACTGGTACCATGG

>Dmel Hairless192-389 WT
CGATGGATCCGAGCAGTGGTTGCAGCAGCAGCTGGCACTGCCAAAATTGGTAAAGGCAGCAA-
CAGCGGTGGCAGTTTTGATATGGGCCGCACACCGATCAGCACCCACGGCAACAATAGTTGGGG
TGGCTATGGCGGCCGTTTACAGTTCTTTAAAGATGGCAAGTTTATTTTAGAACTGGCCCGCAG-
CAAAGATGGCGATAAAAGCGGCTGGGTGAGTGTGACCCGCAAAACCTTTCGCCCGCCGAG
TGCAGCAACCAGCGCAACCGTGACCCCTACCAGTGCCGTGACCACCGCCTACCCGAAGAA

TGAAAACAGCACCTCTTTAAGCTTCAGCGACGACAATAGCAGCATTCAGAGCAGCCCG
TGGCAGCGTGATCAGCCGTGGAAACAGAGTCGTCCGCGCCGTGGCATCAGCAAAGAACTGTC
TTTATTTTTCCACCGCCCGCGCAATAGTACACTGGGTCGTGCAGCCTTACGTACCGCAGCCCG-
CAAACGTCGTCGTCCGCATGAACCGCTGACCACCAGCGAAGATCAGCAGCCGATC
TTTGCCACCGCAATCAAAGCCGAGAACGGTGATGATACTTTAAAAGCCGAAGCAGCCGAATAAC
TGGTACCATGG

>Dmel Hairless192-389 5Amut
CGATGGATCCGAGCCGTTGTGGCAGCAGCAGCTGGCACTGCCAAAATCGGCAAAGGCAGCAA
TAGCGGTGGTAGCTTTGACATGGGCCGCACCCCGATTAGCACCCATGGCAACAACAGCTGGGG
TGGTTATGGTGGTCGTGCCCAAGCTTTTAAAGACGGCAAGTTCATCGCCGAAGCCGCACGCAG-
CAAAGATGGCGACAAAAGCGGTGCCGTGAGCGTGACCCGCAAAACCTTTCGTCCGCCGAG
TGCAGCAACCAGCGCAACCGTTACCCCGACCAGCGCAGTTACCACCGCCTACCCGAAAAAC-
GAAAACAGCACCTCTTTAAGCTTTAGCGACGACAACAGCAGCATTCAGAGCAGCCCG
TGGCAGCGCGATCAGCCGTGGAAACAGAGCCGTCCTCGTCGCGGCATCAGCAAAGAGCTGTC
TTTATTCTTTCATCGCCCGCGCAATAGCACTTTAGGTCGTGCAGCACTGCGCACAGCAGCACG
TAAACGTCGTCGCCCGCATGAACCGCTGACCACCAGCGAAGACCAGCAGCCGA
TTTTTGCCACCGCAATCAAAGCCGAGAACGGCGATGATACTTTAAAAGCAGAAGCAGCCGAA
TAACTGGTACCATGG

>Ptep s-CAP233-432 WT
CGATGGATCCGAACCGTGAATACCGAAGATCCGCCGAAGGATAGCATCAACTTTCTGGACCA-
CAGCCGCGTGACCGATCCGTGTAGTGCCGCAAGCGAAACCAGCCTGCCGCAGGATG
TGCCGGCAACAAGCACCGTGGGCAGCCTGAAATTTTTTCTGGGCGGTCGCCTGGTGCTGAAA
TTAAACGCCCAGCAGGATGGCGGCAGCGGCAATAAATGCCAGTGGGTGCAGAGCAACGATC
TGCCGAAACATAGCAACCATAACAAAAAAGATAAACATAAGAAAAAATTTGCACCGTATAGCTA
TAGCAGCAGCGGCACTCAGAAACCGCTGAAGAAAGGCGACGATACCAGTGCCGTGCCGGACTG
TGATCCGAGCGGCATCAAAAAGCCGCGCCTGAAAGAGTACGAGACCAGCGAGAATAGCGCCC
TGGGTCTGCTGCTGTGCAGCAGCAGTTGGACCCCGCCGGTTGCAGATGGTCAGGAGAGCA
TTGACGTGGACGATACCAGCAGCAAAACCAGCGAGGGCTATATTAGCCCGATCCTGAGCAACAA
TAGCCGCACCAGCAAAATCGACACCATCAAGCACGATTTTGCCAGCAACCCGAACACCTAAC
TGGTACCATGG

>Ptep s-CAP233-432 5Amut
CGATGGATCCGAACCGTGAACACCGAAGACCCGCCGAAAGATAGCATCAACTTTTTAGACCA
TAGCCGCGTGACAGACCCGTGCAGTGCCGCAAGTGAAACCTCTTTACCGCAAGATG
TGCCGGCAACCAGCACCGTGGGTAGCGCCAAAGCCTTTCTGGGCGGTCGTCTGG
TGGCCAAAGCCAATGCCCAGCAAGATGGTGGTAGTGGTAACAAATGCCAAGCTGTGCAGAG-
CAACGATCTGCCGAAACACAGCAATCACAATAAGAAAGACAAACACAAGAAAAAATTTGCCCCG
TATAGCTATAGCAGCAGCGGCACCCAGAAACCGCTGAAAAAAGGCGATGACACCAGCGCAG
TGCCGGATTGCGATCCGAGCGGCATTAAGAAACCGCGTTTAAAGGAGTACGAGACCAGC-
GAAAACAGTGCTTTAGGTTTACTGCTGTGCAGCAGCAGTTGGACACCGCCGGTGGCCGATGG
TCAAGAAGTATCGATGTGGACGACACCAGCAGCAAAACCAGCGAAGGCTACATCAGCCCGA
TTCTGAGCAACAATAGCCGCACCAGCAAAATTGATACCATTAAACATGATTTTGCAAGCAA
TCCGAATACCTAACTGGTACCATGG

## Acknowledgements

We are especially grateful to our many colleagues who made their genome and transcriptome sequence assemblies freely available; without their generosity, this study would not have been possible. We thank Scott Rifkin and Mark Rebeiz for helpful discussion and input during the preparation of the manuscript. We also thank the following artists for making available the illustrations shown in *Figure 2A*: crab, by Firkin (https://openclipart.org/detail/270221/crab-silhouette); monarch butterfly, by carolemagnet (https://openclipart.org/detail/263384/monarch-butterfly); centipede, by Firkin (https://openclipart.org/detail/261126/centipede-3); spider, by liftarn (https://openclipart.org/detail/

179190/spider); horseshoe crab, by Gosc (https://openclipart.org/detail/174556/horseshoe-crab); tick, by Juhele (https://openclipart.org/detail/279073/simple-tick-ixodes-ricinus-silhouette). This work was supported by NIH Grants R01GM046993 and R01GM120377 (to JWP).

## Additional information

### Funding

| Funder | Grant reference number | Author |
|---|---|---|
| National Institute of General Medical Sciences | R01GM046993 | James W Posakony |
| National Institute of General Medical Sciences | R01GM120377 | James W Posakony |

The funders had no role in study design, data collection and interpretation, or the decision to submit the work for publication.

### Author contributions

Steven W Miller, Supervision, Investigation, Visualization, Methodology, Writing—original draft, Writing—review and editing; Artem Movsesyan, Sui Zhang, Investigation, Writing—original draft, Writing—review and editing; Rosa Fernández, Resources, Investigation, Writing—review and editing; James W Posakony, Conceptualization, Data curation, Supervision, Funding acquisition, Investigation, Methodology, Writing—original draft, Writing—review and editing

### Author ORCIDs

Steven W Miller (iD) https://orcid.org/0000-0001-7610-6336
James W Posakony (iD) https://orcid.org/0000-0001-6377-1732

### Decision letter and Author response

Decision letter https://doi.org/10.7554/eLife.48115.020
Author response https://doi.org/10.7554/eLife.48115.021

## Additional files

### Supplementary files

• Supplementary file 1. Representative catalog of Hairless proteins in the Pancrustacea, selected from a curated collection of approximately 400 full-length sequences. Annotated sequences of the entries in this list are provided in *Supplementary file 2*. In the 'MTA microsynteny?' column, + and – in parentheses denote the relative orientation of the *Hairless* and *MTA* genes in the genome. References not in the main reference list are provided in *Supplementary file 6*.
DOI: https://doi.org/10.7554/eLife.48115.012

• Supplementary file 2. Annotated FASTA file of sequences of the Hairless proteins included in *Supplementary file 1*. Characteristic conserved domains and motifs are colored as in *Figure 1A*.
DOI: https://doi.org/10.7554/eLife.48115.013

• Supplementary file 3. Annotated FASTA file of sequences of the Su(H) proteins included in *Table 1*; species for which a full-length Su(H) sequence is not available are omitted here. Colors denote conserved sequence features. Motifs for direct recruitment of the CtBP and Gro co-repressor proteins (aligned in *Table 1*) are shown in red and green, respectively. Large region highlighted in orange is the highly conserved body of Su(H), extending from 'LTREAM' to 'YTPEP'.
DOI: https://doi.org/10.7554/eLife.48115.014

• Supplementary file 4. Representative catalog of S-CAP proteins in the Myriapoda and Chelicerata, selected from a curated collection of approximately 50 full-length sequences. Annotated sequences of the entries in this list are provided in *Supplementary file 5*. In the 'MTA microsynteny?' column, + and – in parentheses denote the relative orientation of the *S-CAP* and *MTA* genes in the genome. The 'H SBD splice?' column indicates whether the Su(H)-binding domain (SBD) in the listed protein is

encoded by two exons with the same splice junction as in *Hairless*. If not, the alternative exon structure is indicated. References not in the main reference list are provided in *Supplementary file 6*.
DOI: https://doi.org/10.7554/eLife.48115.015

• Supplementary file 5. Annotated FASTA file of sequences of the S-CAP proteins included in *Supplementary file 4*. Characteristic domains and motifs are colored as follows: Su(H)-binding domain (SBD), orange; CtBP recruitment motifs, red.
DOI: https://doi.org/10.7554/eLife.48115.016

• Supplementary file 6. Sequence data references not cited in the main paper text.
DOI: https://doi.org/10.7554/eLife.48115.017

• Transparent reporting form
DOI: https://doi.org/10.7554/eLife.48115.018

### Data availability

All data generated or analysed during this study are included in the manuscript and supporting files.

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
