## [Decision Letter]

Thank you for submitting your article "Hairless as a novel component of the Notch signaling pathway" for consideration by *eLife*. Your article has been reviewed by three peer reviewers, and the evaluation has been overseen by Patricia Wittkopp as the Senior and Reviewing Editor. The reviewers have opted to remain anonymous.

The reviewers have discussed the reviews with one another and the Reviewing Editor has drafted this decision to help you prepare a revised submission.

This paper examines the evolution of the Hairless gene and how it became part of the Notch signaling pathway. Solid experimental and computational analyses support the conclusions. It is a revealing case study for understanding how new protein functions evolve.

That said, there was extensive discussion post-review about the major contributions of this work, with concerns raised that the authors were inaccurately claiming to be the first to have discovered Hairless as part of the Notch signaling pathway. My own reading is that this is mostly an issue with word choices in the title and elsewhere in the paper that gave some reviewers this impression that the authors were claiming to have discovered Hairless as a new (novel) component of Notch signaling rather than describing the evolutionary novelty that lead to Hairless being a part of the Notch pathway in some species. For example, I suggest changing the title from "Hairless as a novel component of the Notch signaling pathway" to something like "Evolutionary recruitment of Hairless to the Notch signaling pathway" (although I don't really love "recruitment", as it sounds too active.).

Summary:

Suppressor of Hairless (SuH) is the major transcriptional effector of Notch signals. This study explores the evolutionary origins of Hairless and its mode of interaction with SuH and the two co-repressors Groucho (Gro)and the C-terminal Binding Protein (CtBP). The work persuasively argues that Hairless mediated recruitment of the co-repressors Gro and CtBP to SuH was evolutionarily preceded by direct interactions between SuH and co-repressors. Thus, ancestral direct recruitment of these co repressors, through specific sequences in each protein, evolved into a Hairless mediated (i.e. indirect) recruitment. Hairless proteins seem confined to the Pancrustacea while evidence is presented showing that Myriapods and Chelicerates do harbor a protein with Hairless like "qualities" but not Hairless.

More specifically, outside of the Pancrustacea Su(H) is able to directly recruit transcriptional repressors, which is argued to represent the ancestral phenotype. To support this hypothesis, the authors surveyed many arthropod Su(H) genes to identify repressor recruitment domains, and in so doing identified a novel Gro recruitment motif. In the Pancrustacea, a loss of Su(H)'s ability to directly recruit repressors is linked to the emergence of hairless. The main conclusion of the paper is that hairless evolved from S-CAP proteins found in myriapods and chelicerates. S-CAPs are similar to hairless in that they have a functional Su(H) binding and Groucho and CtBP recruitment motifs. The similarity to Hairless argues that the S-CAP proteins represent an intermediate step in Hairless evolution. This claim is supported by the fact that S-CAPs contain a functional Su(H) binding domain, which is encoded on different exons with exactly the same splice sites as the SBD found in hairless. The authors also identify chelicerate S-CAPs that retain conserved domains found in MTA proteins, and both hairless and S-CAP proteins maintain microsynteny with MTA in a large number of clades. These observations suggest that S-CAP proteins evolved by following tandem duplication of the MTA protein. Their data suggests a mechanism by which a critical component of animal biology (the ability to turn off Notch targets) could adapt a completely novel mechanism (a transition from direct to indirect recruitment of repressors to Su(H)) by tandem duplication and neofunctionalization of one paralog.

Essential revisions:

1) Revise to make it more clear throughout that there is a body of literature already establishing Hairless as part of the Notch signaling pathway, at least in *Drosophila*. Examples of reviewer comments related to this point: "I find the title of the paper somewhat misleading or disingenuous as there is a fairly deep literature establishing Hairless as a component of the Notch signaling pathway." and "The claim in the title, in the impact statement and in the text that Hairless is a novel component of the Notch signaling pathway is incorrect"

2) As one reviewer wrote "The discussion as to why this evolution has occurred and what possible benefits such a transition may have led to (e.g. simultaneous CtBP and Gro binding) is also interesting but still only speculation." and anther wrote "The speculative discussion in Figure 7 is interesting, but it was not clear how likely the putative sources are to being the actual source. Are these the only obvious choices or are there many more? As it stands there is no evidence to argue that these are the sources and it isn't clear that those data should be included without over emphasis on the speculative nature of this figure." Please moderate the Discussion to provide some more context for the likelihood of the particular possibilities described.

3) Please make it more clear that Maier (who has numerous papers related to Hairless evolution) previously reported the existence of a protein with properties like those described in this work.

---

## [Author Response]

[…] There was extensive discussion post-review about the major contributions of this work, with concerns raised that the authors were inaccurately claiming to be the first to have discovered Hairless as part of the Notch signaling pathway. My own reading is that this is mostly an issue with word choices in the title and elsewhere in the paper that gave some reviewers this impression that the authors were claiming to have discovered Hairless as a new (novel) component of Notch signaling rather than describing the evolutionary novelty that lead to Hairless being a part of the Notch pathway in some species. For example, I suggest changing the title from "Hairless as a novel component of the Notch signaling pathway" to something like "Evolutionary recruitment of Hairless to the Notch signaling pathway" (although I don't really love "recruitment", as it sounds to active.).

We wish to make clear that we most certainly were NOT claiming in this paper to be reporting Hairless’s role in the Notch pathway for the first time. We find it very difficult to understand how this interpretation could have gained any traction, in view of the following obvious facts:

• Our lab has a long published history – extending back to 1991 – of investigating the nature and function of the *Hairless* gene and protein as a regulator of cell fate in the fly PNS. And we have made very substantial contributions to defining its role in the Notch pathway as an adaptor protein that mediates the indirect recruitment of two co-repressors to Suppressor of Hairless. As a result, we are of course fully aware of Hairless’s known membership in the pathway. We are utterly mystified how anyone could infer that we were attempting to mislead our colleagues in the field with an obviously false claim.

• The very first figure in our manuscript includes diagrams summarizing for the reader — especially one unfamiliar with the prior studies — Hairless’s function in the Notch pathway. The legend to Figure 1B began as follows in the original version: “Summary of Hairless’s mode of action as an adaptor protein that recruits the global co-repressors C-terminal Binding Protein (CtBP) and Groucho (Gro) to Suppressor of Hairless [Su(H)], the transducing transcription factor for the Notch (N) cell-cell signaling pathway.” Again, it is extremely puzzling how this could be viewed as in any way consistent with an attempt to claim that we were reporting on a new discovery. A summary figure like that would appear last, not first, in a paper describing Hairless’s function for the first time. (Indeed, Figure 1B is based on Figure 6 – the last one – in our 2002 paper in Genes and Development.)

• Our use of the word “novel” in the paper’s title (which seems to have evoked this concern) is fully consistent with standard nomenclature in the field of evolutionary biology. We of course meant to convey that Hairless is an *evolutionary* novelty. Moreover, even a superficial reading of the paper’s Abstract (never mind the paper as a whole) makes it very clear that it is about the *evolution* of Hairless, and not about its current function.

Essential revisions:1) Revise to make it more clear throughout that there is a body of literature already establishing Hairless as part of the Notch signaling pathway, at least in Drosophila. Examples of reviewer comments related to this point: "I find the title of the paper somewhat misleading or disingenuous as there is a fairly deep literature establishing Hairless as a component of the Notch signaling pathway." and "The claim in the title, in the impact statement and in the text that Hairless is a novel component of the Notch signaling pathway is incorrect"

To avoid any possible further confusion, we have made several changes to the manuscript to solidify the reader’s understanding of the topic and intent of our paper.

a) Along the line suggested, we have changed the paper’s title to: “Evolutionary emergence of Hairless as a novel component of the Notch signaling pathway”.

b) We have altered the Impact Statement to read: “Hairless emerged as an evolutionarily novel regulatory protein that replaced an ancestral paradigm of direct co-repressor recruitment by Suppressor of Hairless, its partner transcription factor in the Notch signaling pathway.”

c) We have edited the legend to Figure 1B to start with the phrase “Summary of Hairless’s known mode of action.….”, and have added references to two review articles on Hairless function, including one by Maier.

With regard to other possible wording changes, we have searched the entire manuscript for the words “novel” and “novelty”. Excluding the title and Impact Statement (see above), “novel” appears five times — three times referring to the Gro recruitment motif we found in protostome Su(H)’s, and twice in sentences containing the word “evolutionary” (thus making our intent clear). “Novelty” appears only three times, in each case immediately preceded by the word “evolutionary”. Thus, we see no need for any additional edits (beyond those above) to address this first concern.

2) As one reviewer wrote "The discussion as to why this evolution has occurred and what possible benefits such a transition may have led to (e.g. simultaneous CtBP and Gro binding) is also interesting but still only speculation."

We fully agree that this suggestion is speculative, but we feel that we had already made that quite clear. Please note the following in the relevant part of the original Discussion:

- includes the phrase “…what might have led”….

- includes the phrase “…What capability might it have conferred….”

- begins “One appealing possibility….”

- begins “We suggest that this might be compatible….”

Nevertheless, to be fully explicit about the speculative nature of our suggestion, we have revised the opening sentence of the paragraph referred to as follows:

“One appealing (but of course speculative) possibility is that Hairless may have permitted Su(H) for the first time to recruit both CtBP and Gro simultaneously to the same target genes.”

And another wrote "The speculative discussion in Figure 7 is interesting, but it was not clear how likely the putative sources are to being the actual source. Are these the only obvious choices or are there many more? As it stands there is no evidence to argue that these are the sources and it isn't clear that those data should be included without over emphasis on the speculative nature of this figure." Please moderate the Discussion to provide some more context for the likelihood of the particular possibilities described.

We’re unclear as to the basis for this concern, since the original version of the manuscript contained multiple explicit indicators that we were offering only illustrative possibilities:

- Discussion: “The extensive reconfiguration of this paralog also included the eventual acquisition of the SBD motif and the addition of one or more CtBP recruitment motifs (see Figure 7 for some possible sources of these components).”

- The title of the Figure 7 legend reads as follows: “Speculative possible sources for key elements of the Hairless and S-CAP proteins.”

- The legend to Figure 7B included the following: “A pre-existing gene encoding a protein that utilizes the same PLNLS recruitment motif is a possible source of this exon. Example shown is a portion of the senseless gene….”

We believe the quoted wording had already made it quite clear that we were not claiming to have identified the actual sources of the protein segments referred to.

Still, to make this point even more explicit, we have added the following sentence after the title of the Figure 7 legend:

“Note that these are intended to be only illustrative examples; other sources are of course possible.”

3) Please make it more clear that Maier (who has numerous papers related to Hairless evolution) previously reported the existence of a protein with properties like those described in this work.

We thought we had already done this in the original version of the manuscript. Please note that the Results section explicitly included the following sentence: “We note that the occurrence of this protein in the centipede *Strigamia maritima* has also recently been reported by Maier (Maier, 2019).” Then, in the legend to Figure 4A (the sequence alignment panel illustrating the S-CAP similarities to the Hairless SBD), we again cited Maier’s paper in connection with the centipede protein.

However, to clarify this point further, we have now added a sentence to the Figure 4A legend crediting Maier more explicitly:

“We note that Maier (Maier, 2019) has previously described the presence of the SBD-like element in the *Strigamia maritima* sequence.”